# A 20-Year Analysis of the Dynamics and Driving Factors of Grassland Desertification in Xilingol, China

**Jingbo Li** [1,2], **Chunxiang Cao** [1,2,*], **Min Xu** [1,2], **Xinwei Yang** [1], **Xiaotong Gao** [1,2], **Kaimin Wang** [1,2], **Heyi Guo** [1,2] and **Yujie Yang** [1,2]

1   State Key Laboratory of Remote Sensing Science, Aerospace Information Research Institute, Chinese Academy of Sciences, Beijing 100094, China; lijingbo21@mails.ucas.ac.cn (J.L.); xumin@radi.ac.cn (M.X.); yangxw@aircas.ac.cn (X.Y.); gaoxiaotong21@mails.ucas.ac.cn (X.G.); wangkaimin19@mails.ucas.ac.cn (K.W.); guoheyi20@mails.ucas.ac.cn (H.G.); yangyujie22@mails.ucas.ac.cn (Y.Y.)
2   University of Chinese Academy of Sciences, Beijing 101408, China
*   Correspondence: caocx@aircas.ac.cn

**Abstract:** Grassland desertification stands as an ecological concern globally. It is crucial for desertification prevention and control to comprehend the variation in area and severity of desertified grassland (DGL), clarify the intensities of conversion among DGLs of different desertification levels, and explore the spatial and temporal driving factors of desertification. In this study, a Desertification Difference Index (DDI) model was constructed based on albedo-EVI to extract desertification information. Subsequently, intensity analysis, the Geo-detector model, and correlation analysis were applied to analyze the dynamics and driving factors of desertification. The results showed the following: (1) Spatially, the DGL in Xilingol exhibited a zonal distribution. Temporally, the degree of DGL decreased, with the proportion of severely and moderately desertified areas decreasing from 51.77% in 2000 to 37.23% in 2020, while the proportion of nondesertified and healthy areas increased from 17.85% in 2000 to 37.40% in 2020; (2) Transition intensities among different desertification levels were more intense during 2000–2012, stabilizing during 2012–2020; (3) Meteorological factors and soil conditions primarily drive the spatial distribution of DDI, with evapotranspiration exhibiting the most significant influence (*q*-value of 0.83), while human activities dominate interannual DDI variations. This study provides insights into the conversion patterns among different desertification levels and the divergent driving forces shaping desertification in both spatial and temporal dimensions in Xilingol.

**Keywords:** Albedo–EVI; desertification; intensity analysis; Geo-detector; driving factors; correlation analysis





## 1. Introduction

Desertification denotes the degradation of land in arid, semiarid, and semihumid regions, resulting from a combination of factors including climate change and human activities [1]. With social development and climate change, desertification has become an increasingly severe issue and is now one of the most serious challenges faced by humanity [2]. Statistics showed that area of desertification was approximately $3.6 \times 10^7$ km$^2$ in 2009 globally, constituting 25% of the Earth's total land area. Currently, land desertification impacts two-thirds of the world's nations and nearly 20% of the global population [3]. In the 21st century, land desertification has emerged as a major environmental challenge, significantly hindering the prospects of sustainable development [4]. China stands as one of the countries most significantly affected by desertification on a global scale, experiencing the largest land area impacted, the largest population affected, and the most severe wind and sand hazards. Presently, the extent of desertified land in China encompasses $1.72 \times 10^6$ km$^2$, representing 17.93% of the entire national land area. Desertification is distributed across 920 counties in 30 provinces (autonomous regions or municipalities) [5].

The top five provincial-level administrative regions with the largest desertification areas are Xinjiang, Inner Mongolia, Tibet, Qinghai, and Gansu. Among these regions, Inner Mongolia's desertification area comprises 23.7% of the national desertified land, positioning it as one of the areas most heavily impacted by desertification [6]. Furthermore, Inner Mongolia's grassland, as China's largest grassland, plays a crucial role as a significant livestock production base. The management and protection of grasslands are of paramount importance for maintaining China's ecological security [7]. Consequently, the control of desertified grassland (DGL) has garnered widespread attention from the government and various sectors of society.

Confronted with the escalating challenge of grassland desertification, precise, timely, and efficacious monitoring of DGL assumes paramount significance in comprehending the dynamics of the desertification processes and devising strategies for its prevention and control [8]. Field surveys were once the main method for monitoring vegetation changes but were challenging to conduct over long time periods and large spatial scales. Additionally, they required substantial human, financial, and material resources [9]. The application of remote sensing has provided a new possibility for continuous monitoring of DGL over extended periods and large spatial extents [10,11].

At present, the extraction of remote sensing information for desertification primarily relies on visual interpretation or automatic classification using machine learning methods [12]. This process is integral to the monitoring, mapping, analysis, and assessment of desertification. Traditional visual interpretation methods are not only susceptible to human factors but also entail high work intensity and low efficiency. The application of automatic classification methods in large-scale regions is constrained by limitations on classification accuracy. Simultaneously, conventional remote sensing information extraction methods exhibit a low level of utilization, thereby impeding the optimal contribution of abundant remote sensing information to desertification monitoring. Zeng et al. introduced the Desertification Difference Index (DDI) model [13], which utilizes the albedo-NDVI feature space to streamline and improve the monitoring of desertification on a large scale. This model, characterized by easy data acquisition and good temporal and spatial continuity, resolved the problem of data accessibility in most desertification index systems. Therefore, it has been widely adopted by researchers [14–19].

However, due to nonlinear stretching, NDVI suppresses high-value areas, making it prone to saturation [20]. Furthermore, NDVI only considers the influence of near-infrared and red bands, failing to effectively control the impact of atmospheric and soil factors on vegetation reflectance spectra. The Enhanced Vegetation Index (EVI) has improved and optimized NDVI, further mitigating the effects of atmospheric and soil factors [21,22]. Moreover, existing studies predominantly explore the drivers of desertification from either a temporal or spatial dimension alone, with limited comprehensive analyses of desertification drivers from both temporal and spatial perspectives. Thus, this study employed the EVI to construct the DDI and comprehensively investigated the drivers of desertification in the Xilingol grassland from both spatial distribution and temporal change perspectives. This approach aims to provide the public with a clearer and more comprehensive understanding.

The factors influencing desertification are complex and diverse. Studies have confirmed the substantial influence of meteorological factors on land desertification, including temperature, precipitation, wind speed, and evapotranspiration [14,23–28]. Regions prone to desertification generally experience large temperature differences, scarce precipitation, and high wind speeds, which directly or indirectly lead to sparse vegetation and fragile ecological environments, hindering effective desertification prevention [29]. Apart from meteorological factors, various environmental factors such as different topographies, vegetation cover types, soil types, and surface soil moisture levels can also influence the occurrence of land desertification [30–32]. Moreover, human activities, including overgrazing, intensive cultivation, and extensive industrial land development, significantly contribute to land desertification [33–35].

This study employed intensity analysis and a geographical detector model to investigate the dynamics of desertification and the spatial distribution drivers of desertification. In previous research, land-use dynamic models and transition matrices have commonly been used to elucidate desertification dynamics [36–38]. However, these methods require the separate determination of mutual conversions between different land categories using transition matrices, which hinders a systematic and quantitative evaluation of desertification dynamics. In contrast, intensity analysis demonstrates significant advantages in terms of systematic and quantitative assessments, making it widely utilized by researchers [39–43]. The Geo-detector model, a statistical method for exploring spatial heterogeneity and its driving factors, not only investigates the individual impact of a single factor on desertification but also detects the interaction effects between two variables influencing desertification. This model is widely utilized in research pertaining to economics, population dynamics, land use, and other fields [26,39–41,43].

This study aimed to do the following: (1) quantify the variations in DGL area and analyze the spatial patterns of DGL, (2) elucidate the processes of change and conversion patterns in the DGLs of different levels, and (3) explore and compare the driving factors of the DGL in Xilingol from 2000 to 2020 from both temporal and spatial dimensions. This study holds significant value in enhancing our understanding of the spatial distribution characteristics, driving factors, and interactions related to grassland desertification. It provides valuable insights for subsequent research and desertification prevention efforts.

## 2. Materials and Methods

### 2.1. Study Area

Xilingol League is located in central Inner Mongolia, spanning from 41.4°N to 46.6°N and from 111.1°E to 119.7°E (Figure 1), with a total area of 206,000 km$^2$. The region displays a temperate continental climate characterized by arid and semiarid features, including an average annual temperature of 2.2 °C, approximately 280 mm of average annual precipitation, and elevations ranging from 760 m to 1925 m above sea level. The terrain is predominantly plateau, sloping from south to north, featuring low hills and basins in the eastern and southern sectors. A distinct zonal pattern is observed in soil and vegetation coverage, with soils transitioning from black calcareous soil to light and dark chestnut soils from southeast to northwest. This transition aligns with noticeable variations in vegetation coverage, progressing from forests to meadow steppe, typical steppe, and desert steppe. Grasslands, covering around 87% of the entire region, dominate the landscape [43–45].

Xilingol League comprises twelve county-level administrative regions, with a total resident population of 1.1071 million. The twelve banners and counties include Xilinhot City (XC), Erenhot City (EC), East Ujimqin Banner (EUB), West Ujimqin Banner (WUB), Abag Banner (AB), Sunite Left Banner (SLB), Sunite Right Banner (SRB), Xianghuang Banner (XHB), Zhengxiangbai Banner (ZXBB), Zhenglan Banner (ZLB), Taibus Banner (TB), and Duolun County (DLC) [43].

### 2.2. Data and Processing

The data used in this study can be categorized into six types, including vegetation index, reanalysis meteorological data, soil data, topographical data, human activities data, and statistics data. The specific data sources and details are provided in Table 1. Regarding the EVI data, the study utilizes the annual maximum values, while precipitation is represented by the annual cumulative values. Surface temperature, wind speed, evapotranspiration, surface albedo, and soil moisture are all represented by annual averages. For population density, the average values for the years 2000–2020 were calculated. Spatial distribution data for livestock density were available only for the years 2020 and 2015; hence, the livestock density data used in this study represent the average for these two years. Additionally, population, livestock numbers (end of June) and GDP statistics for Xilingol from 2000 to 2020 were obtained by referring to the Xilingol Statistical Yearbook.

These data have been well applied in relevant studies conducted in the Inner Mongolia region [14,26,40,43,44,46].

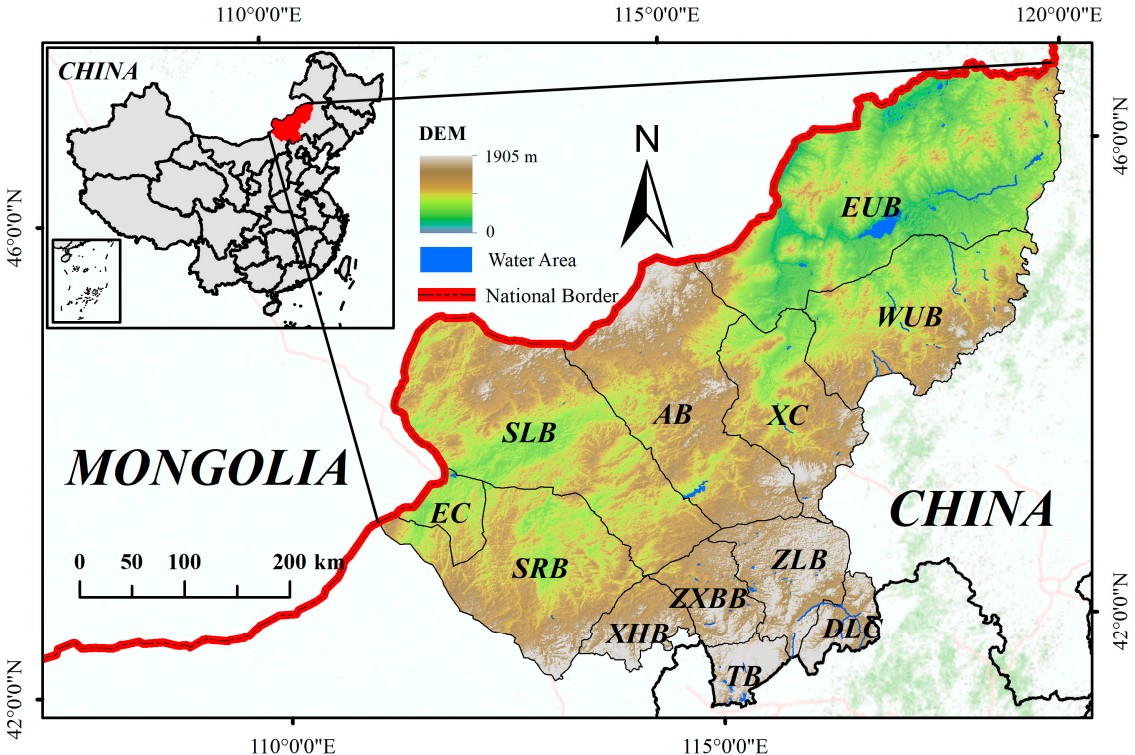

**Figure 1.** Location of study area. The upper-left map displays the location of Xilingol within China, while the main map illustrates the topography and administrative subdivisions of Xilingol.

**Table 1.** Data description and sources.

| Category | Name | Code | Source | Spatial Resolution | Temporal Resolution | Unit |
|---|---|---|---|---|---|---|
| Vegetation index | Enhanced Vegetation Index | EVI | https://lpdaac.usgs.gov/ (accessed on 4 September 2023) | 250 m | 16 day | / |
| Meteorological factors | Land surface temperature | LST | https://lpdaac.usgs.gov (accessed on 4 September 2023) | 1000 m | Per day | °C |
| | Precipitation | PRE | https://chc.ucsb.edu/data | 0.05° | Per day | mm |
| | Wind velocity | WV | https://www.climatologylab.org/terraclimate.html (accessed on 4 September 2023) | 4638.3 m | Per month | m/s |
| | Evapotranspiration | ET | https://lpdaac.usgs.gov/ (accessed on 4 September 2023) | 500 m | 8 day | kg/m$^2$ |
| | Albedo | / | https://lpdaac.usgs.gov/ (accessed on 4 September 2023) | 500 m | Per day | / |
| Soil factors | Soil type | ST | http://www.resdc.cn (accessed on 5 September 2023) | shapefile | / | / |
| | Soil erosion type | SET | http://www.resdc.cn (accessed on 6 September 2023) | shapefile | / | / |
| | Soil erosion intensity | SEI | http://www.resdc.cn (accessed on 5 September 2023) | shapefile | / | / |
| | Soil moisture | SM | https://disc.gsfc.nasa.gov (accessed on 5 September 2023) | 27,830 m | / | kg/m$^2$ |
| Topographical factors | Digital elevation model | DEM | https://cmr.earthdata.nasa.gov/ (accessed on 4 September 2023) | 30 m | / | m |
| | Slope | SLP | Calculated from DEM | 30 m | / | ° |
| | Aspect | ASP | Calculated from DEM | 30 m | / | / |

**Table 1.** *Cont.*

| Category | Name | Code | Source | Spatial Resolution | Temporal Resolution | Unit |
|---|---|---|---|---|---|---|
| Human activities | Population | POP | https://sedac.ciesin.columbia.edu/ (accessed on 10 September 2023) | 1000 m | Per year | persons/km$^2$ |
| | Livestock | LIV | https://www.fao.org/ (accessed on 10 September 2023) | 10,000 m | / | heads/km$^2$ |
| | Land use/cover change | LUCC | https://lpdaac.usgs.gov/ (accessed on 10 September 2023) | 500 m | Per year | / |
| Statistics | Population | / | http://tjj.xlgl.gov.cn/ (accessed on 11 September 2023) | / | Per year | k |
| | GDP | / | http://tjj.xlgl.gov.cn/ (accessed on 11 September 2023) | / | Per year | billion RMB |
| | Livestock | / | http://tjj.xlgl.gov.cn/ (accessed on 12 September 2023) | / | Per year | million |

To ensure consistency in spatial resolution for input data, all geographical spatial data in this study were resampled to 250 m. In terms of statistical data, a normalization process was applied to the acquired data.

*2.3. Methods*

The workflow of this study comprised five main components (Figure 2). Firstly, data input, which encompassed topographic data, vegetation index, geography-based human activities data, soil data, data from statistical yearbooks, and reanalysis meteorological data. Secondly, data preprocessing was conducted, involving reprojection, resampling, cropping, calculation, and standardization. Thirdly, the Enhanced Vegetation Index (EVI) and surface albedo were utilized to construct the DDI, with subsequent threshold delineation of the DDI results. Finally, based on the DDI and threshold delineation outcomes, the study employed intensity analysis, the geographical detector model, and correlation analysis to analyze the spatiotemporal dynamics and drivers of desertification in the Xilingol grassland.

2.3.1. Desertification Difference Index (DDI) Model Construction

According to the DDI model proposed by Zeng et al. [13], albedo-NDVI exhibit a significant negative linear correlation in a one-dimensional feature space [13,15–18]. In this study, the model was enhanced by replacing NDVI with EVI (Figure 3), effectively reducing the impact of atmosphere and soil. A higher DDI value indicates a lower degree of desertification. The basic equations for this modified model are presented in Equations (1)–(3):

$$DDI = \alpha \cdot EVI\text{-}Albedo \tag{1}$$

$$Albedo = k \cdot EVI + b \tag{2}$$

$$\alpha \, k = 1 \tag{3}$$

In the equations, k is the slope, b is the constant term, and $\alpha$ is the reciprocal of k.

Researchers tend to employ the natural breakpoint method to classify DDI values after calculation of DDI [14,18]. While this approach is feasible when analyzing data for a specific year, it lacks meaningful implications for long-term data comparisons or trend analyses. Therefore, in this study, a uniform threshold for desertification classification was determined based on human–machine interaction, as shown in Table 2. Subsequently, 200 random points within the study area were selected, and the DDI results for the years of 2006, 2012, and 2020 were validated using Google Earth historical images. The accuracy rates were 82%, 85.5%, and 80% respectively, meeting the experimental requirements.

**Input data**

**Topographic Data**
- DEM

**Vegetation index**
- EVI

**Geography-based Human Activities Data**
- Population
- Livestock
- LUCC

**Soil Data**
- Soil Type
- Soil Erosion Type
- Soil Erosion Intensity
- Soil Moisture

**Data from Statistical Yearbook**
- Population
- Livestock
- GDP

**Reanalysis Meteorological Data**
- Land surface Temperature
- Precipitation
- Wind Velocity
- Evapotranspiration
- Albedo

**Data Preprocessing**

Reprojection　Resampling　Clipping　Calculation　Standardization

**Model Construction**

Albedo + EVI → Desertification Difference Index → Threshold Division

**Analysis**

*Spatiotemporal Variation*
- Results based on Threshold Division
- Area Statistics
- Spatiotemporal Variation of Desertification at Different Levels

*Intensity Analysis*
- Transition Matrix
- Intensity Analysis
- Interval Level
- Category Level
- Transition Level

*Driving Factors*
- DDI
- Meteorological Factors
- Soil Factors
- Topographic Factors
- Geography-based Human Activities Factors
- Geo-detector
- Factor Detector
- Ecological Detector
- Interaction Detector
- Driving factors of DDI Spatial Distribution

- DDI
- Meteorological Factors
- Statistic-based Human Activities Factors
- Correlation Analysis
- Driving Factors Of DDI Temporal Variation

**Figure 2.** Workflow of the study. The study consisted of four parts: input data, data preprocessing, model construction, and analysis.

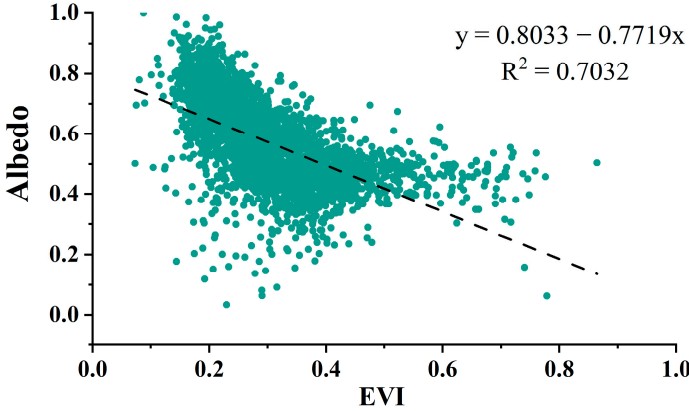

$$y = 0.8033 - 0.7719x$$
$$R^2 = 0.7032$$

**Figure 3.** Linear fitting of albedo-EVI (the figure shows the result from 2020).

**Table 2.** Classification standards for desertification level.

| Category | Value of DDI |
|---|---|
| I (Severely Desertified) | DDI < −0.45 |
| II (Moderately Desertified) | −0.45 ≤ DDI < −0.25 |
| III (Slightly Desertified) | −0.25 ≤ DDI < −0.05 |
| IV (Nondesertified) | −0.05 ≤ DDI < 0.15 |
| V (Healthy) | DDI ≥ 0.15 |

### 2.3.2. Data Normalization and Weighting

To explore the driving forces of temporal variations in DDI, factors were categorized into two groups, and normalization was performed on each group. Assuming that the impact weights of each factor on the composite value in both groups are equal, in this study, we employed the method of average weighting to calculate the composite values for meteorological and human activity factors, respectively. The calculation is formulated as shown in Equations (4) and (5):

$$x_n = \frac{x_i - x_{min}}{x_{max} - x_{min}} (i = 1, 2, \ldots, n) \tag{4}$$

$$X = \frac{1}{n}(x_1 + x_2 + \ldots + x_n) \tag{5}$$

In the equations, $x_n$ is the normalized value of each factor, $x_i$ is the original value, $x_{min}$ is the minimum value, $x_{max}$ is the maximum value, and $X$ is the composite value of each group.

### 2.3.3. Spearman Correlation Coefficient

The study employed the Spearman correlation coefficient to analyze the correlation between variables. DDI averages for each two-year period from 2000 to 2020 were computed, with values for intervening years derived as the mean of adjacent two-year DDI values. To assess the predominant influence of meteorological factors versus human activities on DDI changes, separate weighted averages were calculated for both sets of factors. Subsequent testing revealed nonnormal distributions for both sets of values, leading to the selection of the Spearman correlation coefficient for correlation analysis [47,48]. These were calculated according to Equations (6) and (7):

$$\rho = 1 - \frac{6 \sum d_i^2}{n(n^2 - 1)} \tag{6}$$

$$d_i = x_i - y_i \tag{7}$$

In these equations, $\rho$ is the Spearman correlation coefficient, $n$ is the sample size, and $x_i$ and $y_i$ are the variables.

### 2.3.4. Intensity Analysis

The intensity analysis method is primarily employed for quantitatively analyzing variations in changes among diverse land categories [42,49]. Intensity analysis involves three levels: interval level, category level, and transition level [42].

The mathematical notations used in intensity analysis include the following: $i$, representing a category at the initial time point within a specific time period; $j$, representing a category at the final time point within a specific time period; $J$, denoting the total number of categories; $m$, representing the losing category in a specified transition; $n$, representing the gaining category in a specified transition; $T$, denoting the number of time point; and $t$, representing the initial time point within $[Y_t, Y_{t+1}]$, with $t$ ranging from *1* to $T - 1$.

1. Interval level

Two indicators are introduced at this level: $S_t$, representing the annual percentage of actual changed area over the total area within each time interval, denoting the average annual change rate over $T - 1$ intervals; and U, representing the annual percentage of theoretical changed area over the total study area per time interval, spanning the entire study period. The terms "slow" and "fast" were employed to characterize the relative rates of change within these time intervals. These were calculated according to Equations (8) and (9):

$$S_t = \frac{\left\{ \sum_{j=1}^{J} \left[ \left( \sum_{i=1}^{J} c_{tij} \right) - C_{tii} \right] \right\} / \left[ \sum_{j=1}^{J} \left( \sum_{i=1}^{J} C_{tij} \right) \right]}{Y_{t+1} - Y_t} \times 100\% \tag{8}$$

$$U = \frac{\sum_{t=1}^{T-1} \left\{ \sum_{j=1}^{J} \left[ \left( \sum_{i=1}^{J} c_{tij} \right) - C_{tii} \right] \right\} / \left[ \sum_{j=1}^{J} \left( \sum_{i=1}^{J} C_{tij} \right) \right]}{Y_T - Y_1} \times 100\% \tag{9}$$

2. Category level

At this level, two indicators are employed. Firstly, the reduction intensity $L_{ti}$ signifies the annual percentage of land class *i* area reduced from the initial total area of class *i* during the study period $[Y_t, Y_{t+1}]$. Secondly, the increase intensity $G_{tj}$ of each land class *j* represents the annual percentage of area added to class *j* from the final total area of class *j* within the study period $[Y_t, Y_{t+1}]$, indicating changes from non-*j* land use types to land class *j*. $L_{ti}$ and $G_{tj}$ are calculated according to Equations (10) and (11).

$$L_{ti} = \frac{\left[ \left( \sum_{j=1}^{J} c_{tij} \right) - C_{tii} \right] / (Y_{t+1} - Y_t)}{\sum_{j=1}^{J} C_{tij}} \times 100\% \tag{10}$$

$$G_{tj} = \frac{\left[ \left( \sum_{i=1}^{J} C_{tij} \right) - C_{tjj} \right] / (Y_{t+1} - Y_t)}{\sum_{i=1}^{J} C_{tij}} \times 100\% \tag{11}$$

3. Transition level

At this level, four change indicators are derived from two perspectives. Firstly, an increase in land class *n* is assessed using observed transition intensities $R_{tin}$ from land class *i* to land class *n*, where $J - 1$ transfer intensities exist from other land classes to land class *n* within each time interval. The average increase intensity $W_{tn}$ of land class *n* is calculated, indicating the average transfer intensity from other land classes to land class *n*, weighted by their respective areas. Secondly, a decrease in land class *m* is evaluated through observed transition intensities $Q_{tmj}$ from land class *m* to land class *j*. The average decrease intensity $V_{tm}$ of land class *m* is computed. $R_{tin}$, $W_{tn}$, $Q_{tmj}$, and $V_{tm}$ are calculated according to Equations (12)–(15):

$$R_{tin} = \frac{\frac{C_{tin}}{(Y_{t+1} - Y_t)}}{\sum_{j=1}^{J} C_{tij}} \times 100\% \tag{12}$$

$$W_{tn} = \frac{\frac{\left[ \left( \sum_{i=1}^{J} C_{tin} \right) - C_{tnn} \right]}{(Y_{t+1} - Y_t)}}{\sum_{j=1}^{J} \left[ \left( \sum_{i=1}^{J} C_{tij} \right) - C_{tnj} \right]} \times 100\% \tag{13}$$

$$Q_{tmj} = \frac{\frac{C_{tmj}}{(Y_{t+1} - Y_t)}}{\sum_{i=1}^{J} C_{tij}} \times 100\% \tag{14}$$

$$V_{tm} = \frac{\frac{\left[\left(\sum_{j=1}^{J} C_{tmj}\right) - C_{tmm}\right]}{(Y_{t+1} - Y_t)}}{\sum_{i=1}^{J}\left[\left(\sum_{j=1}^{J} C_{tij}\right) - C_{tim}\right]} \times 100\% \tag{15}$$

### 2.3.5. Geographical Detector

The geographical detector is a powerful tool for conducting driver and factor analyses [50,51]. The model comprises four detectors: risk detector, factor detector, ecological detector, and interaction detector. In this study, the factor detector, ecological detector, and interaction detector were employed [42].

1.  Factor Detector

The $q$-value (ranging from 0 to 1) quantifies how well factor $x$ explains the spatial variation of attribute $y$. A higher $q$-value indicates stronger explanatory power, determined through a significant F-test.

$$q = 1 - \frac{\sum_{h=1}^{L} N_h \sigma_h^2}{N \sigma^2} \tag{16}$$

Within the equation, "$h = 1 \ldots L$" signifies distinct strata or layers for both the independent variable $x$ and the dependent variable $y$, essentially representing categories or partitions. Additionally, $N_h$ stands for the quantity of units in layer $h$, while $N$ denotes the total number of units across the entire region. Furthermore, $\sigma_h^2$ represents the variance of $y$ values within layer $h$, and $\sigma^2$ denotes the overall variance of $y$ values across the entire region.

2.  Ecological Detector

Ecological detector is employed to determine if notable distinctions exist in the spatial distribution impacts on attribute $y$ between two factors. This evaluation is quantified through the $F$-statistic. These are calculated according to Equations (17) and (18):

$$F = \frac{N_{x_1}(N_{x_2} - 1)\text{SSW}_{x_1}}{N_{x_2}(N_{x_1} - 1)\text{SSW}_{x_2}} \tag{17}$$

$$\text{SSW}_{x_1} = \frac{\sum_{h=1}^{L_1} N_h \sigma_h^2}{N \sigma^2}, \text{SSW}_{x_2} = \frac{\sum_{h=1}^{L_2} N_h \sigma_h^2}{N \sigma^2} \tag{18}$$

In these equations, $N_{x_1}$ and $N_{x_2}$ are the number of factors for $x_1$ and $x_2$, respectively; $SSW_{x_1}$ and $SSW_{x_2}$ are the accumulation of squares within each grouping established by the two factors; and $L_1$ and $L_2$ signify the count of classes or categories for $x_1$ and $x_2$, respectively.

3.  Interaction Detector

Interaction detector assesses interactions between risk factors $x$, determining whether $x_1$ and $x_2$ jointly influence the explanatory power of dependent variable $y$, or if their effects on $y$ are independent. It calculates $q$-values for $x_1$ and $x_2$ ($q(x_1)$ and $q(x_2)$) and the interaction q-value ($q(x_1 \neq x_2)$) formed by the interplay of $x_1$ and $x_2$. Comparing $q(x_1 \neq x_2)$ with $q(x_1)$ and $q(x_2)$ reveals the interaction type. Further details on interaction detector can be found in Table 3.

**Table 3.** Definition of interaction detector.

| Description | Interaction |
| --- | --- |
| $P(x_1 \cap x_2) < \min(P(x_1), P(x_2))$ | Nonlinear weaken |
| $\min(P(x_1), P(x_2)) < P(x_1 \cap x_2) < \max(P(x_1), P(x_2))$ | Uni-weaken |
| $P(x_1 \cap x_2) > \max(P(x_1), P(x_2))$ and $P(x_1 \cap x_2) < P(x_1) + P(x_2)$ | Bi-enhance |
| $P(x_1 \cap x_2) > P(x_1) + P(x_2)$ | Nonlinearly enhance |
| $P(x_1 \cap x_2) = P(x_1) + P(x_2)$ | Independent |

## 3. Results

### 3.1. Analysis of DDI Distribution and Spatiotemporal Trends

3.1.1. Spatiotemporal Distribution Characteristics of DDI

In this study, we calculated the DDI value for Xilingol every two years, and the values were categorized into five groups (Figure 4). Spatially, desertification levels exhibited a systematic east-to-west intensification, aligning with the distribution of meadow steppe, typical steppe, and desert steppe in Xilingol (Figure 5). Temporally, proportions of DGL levels were calculated for each period (Figure S1). It was observed that the area of level I showed a fluctuating decreasing trend, declining from 22.03% in 2000 to 15.15% in 2020, and the proportion of level II areas decreased from 29.74% in 2000 to 22.08% in 2020. In contrast, the areas of level IV and level V increased annually. The combined proportion of these two levels rose from 17.85% in 2000 to 37.40% in 2020, and the combined area of the two was more than double the initial total. These results indicate an amelioration in the degree of grassland desertification over the course of 20 years.

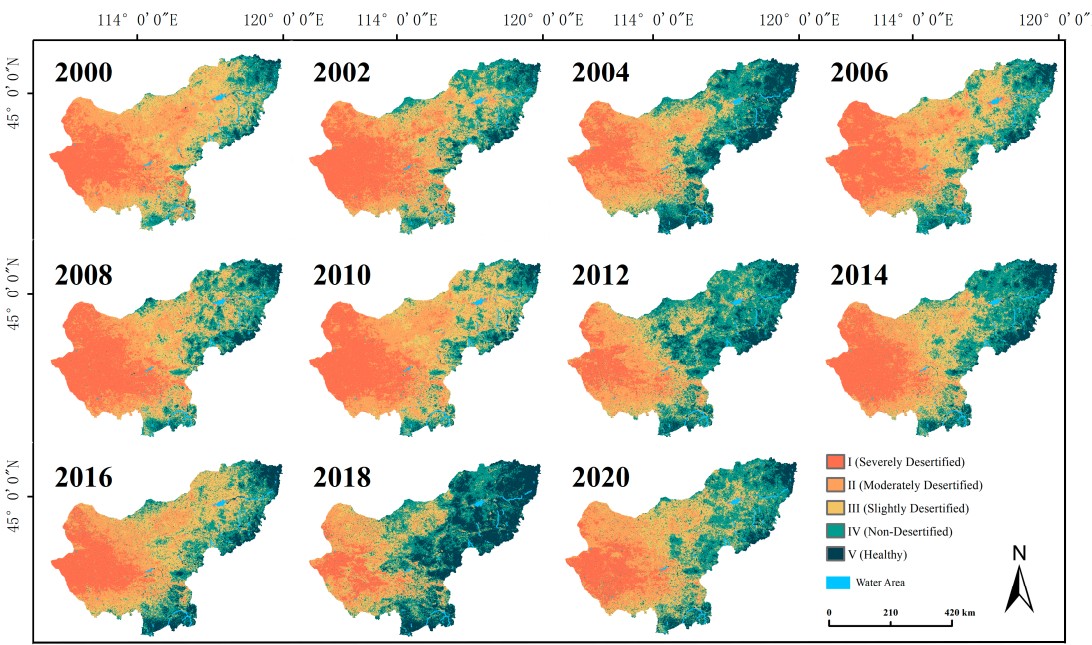

**Figure 4.** Spatiotemporal distribution of grassland desertification degree in Xilingol.

3.1.2. Spatiotemporal Variation of DGL

A statistical analysis was conducted on areas exhibiting various degrees of desertification (Table S1). From 2000 to 2010, level I increased from 44,518.60 km$^2$ to 56,903.04 km$^2$, indicating a worsening trend, particularly intensifying from 2004 to 2010. During this period, levels I, II, and III expanded from 28,395.93 km$^2$, 44,344.86 km$^2$, and 39,155.07 km$^2$ in 2004 to 56,903.04 km$^2$, 48,520.86 km$^2$, and 55,841.82 km$^2$ in 2010, respectively.

Contrastingly, from 2010 to 2020, desertification showed a decline. The most substantial decrease occurred in level I, decreasing from 56,903.04 km$^2$ to 30,606.31 km$^2$. Simultaneously, levels IV and V significantly increased from 29,872.40 km$^2$ and 10,912.78 km$^2$ in 2010 to 54,934.55 km$^2$ and 20,633.46 km$^2$ in 2020, respectively.

Across 2000–2020, levels I, II, and III areas decreased, while levels IV and V increased significantly. Notably, level IV expanded from 27,780.69 km$^2$ in 2000 to 53,273.86 km$^2$ in 2020, a 91.78% increase. Level V grew from 8281.21 km$^2$ to 22,428.92 km$^2$ in 2020, nearly tripling its original size.

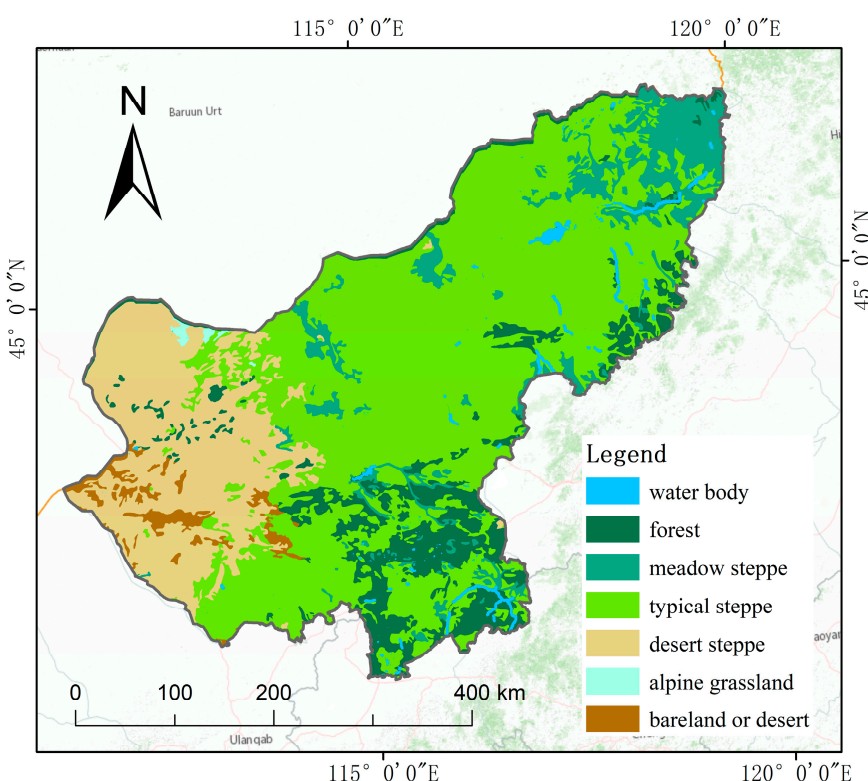

**Figure 5.** Spatial distribution of vegetation in Xilingol.

Linear fitting was employed to observe changes in different degrees of grassland desertification (DGL) areas (Figure 6). Notably, levels I, II, and III decreased, while Levels IV and V increased. Examining the absolute values of the fitted lines' slopes revealed change rates, which ranked as follows: level V (1115.49) > level I (1109.25) > level IV (832.93) > level II (620.98) > level III (217.66). Consequently, it can be inferred that from 2000 to 2020, level V increased the fastest, while level I decreased the fastest, indicating an improvement in the desertification situation of Xilingol.

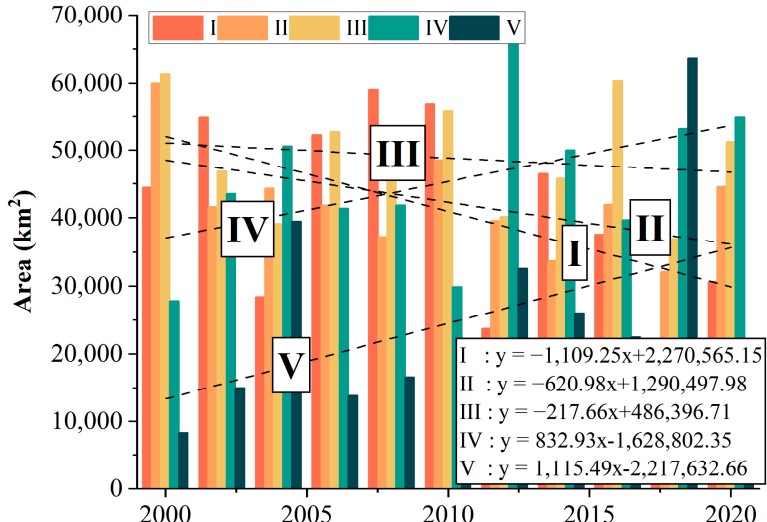

**Figure 6.** Linear fitting for the area variation of different levels of DGL.

*3.2. Intensity Analysis*

In this study, based on the years of DDI calculation and to accurately capture the dynamics of desertification changes, the period from 2000 to 2020 was divided into five

cycles for intensity analysis. Intensity analysis comprises three levels: interval level, category level, and transition level.

### 3.2.1. Interval Level

At the interval level (Figure 7), the sections to the left and right of the zero value depict the proportion of the total area changed over the cycle and the proportion of average annual change area during the cycle, respectively. This illustrates the overall conversion intensity among different desertification levels within a specific period. The red dashed line serves as a uniform intensity indicator, representing the overall change intensity from 2000 to 2020; values surpassing this threshold indicate fast conversion, while values below it suggests slow conversion.

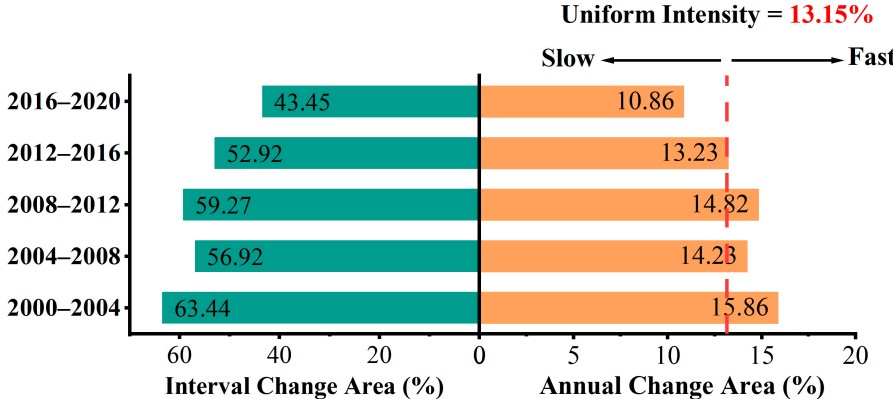

**Figure 7.** Interval level of intensity analysis.

It is evident that the intensity during the periods 2000–2004, 2004–2008, and 2008–2012 significantly exceeded the uniform value. Consequently, during these three periods, there were drastic changes among different desertification levels, with the period from 2000 to 2004 exhibiting the highest intensity, while the intensity was relatively mild from 2016 to 2020.

### 3.2.2. Category Level

At the category level (Figure 8), the values to the left of zero represent the increase in area from other levels of DGL converted to the specific level during the period and the decrease in area from the specific level of DGL converted to other levels during the period. The values to the right of zero represent the average annual gain and loss intensity for the specific level of DGL during the period. The red dashed line denotes the uniform intensity, determined by the average annual change intensity S from the interval level. If a bar surpasses the uniform line, it indicates a relatively active change in the DGL category during the specified time interval; otherwise, the change is relatively steady.

It can be observed that during the 2000–2004 period (Figure 8A), the loss intensities for levels III and IV exceeded the average, and the gain intensities for levels III, IV, and V surpassed the average. This suggests intense conversions among levels III, IV, and V during this period. In the 2004–2008 period (Figure 8B), the gain and loss intensities for almost all desertification levels were close to or greater than the average, making it one of the most active periods for conversions among DGL categories. In the 2008–2012 period (Figure 8C), the gain intensities for levels IV and V were significantly greater than their respective loss intensities, while the loss intensity for level I was significantly higher than the gain intensity. This indicates a notable improvement in the desertification situation during this period. In the 2012–2016 period (Figure 8D), the gain intensities of levels I, II, and III were greater than the loss intensities, while the loss intensities for levels IV and V were greater than the gain intensities, suggesting an exacerbation of desertification during this period. In the 2016–2020 period (Figure 8E), the loss intensities for levels I and III were

greater than the gain intensity, and only the gain intensity for level IV and the loss intensity for level III significantly exceeded the average, indicating a stable and positive trend during this period.

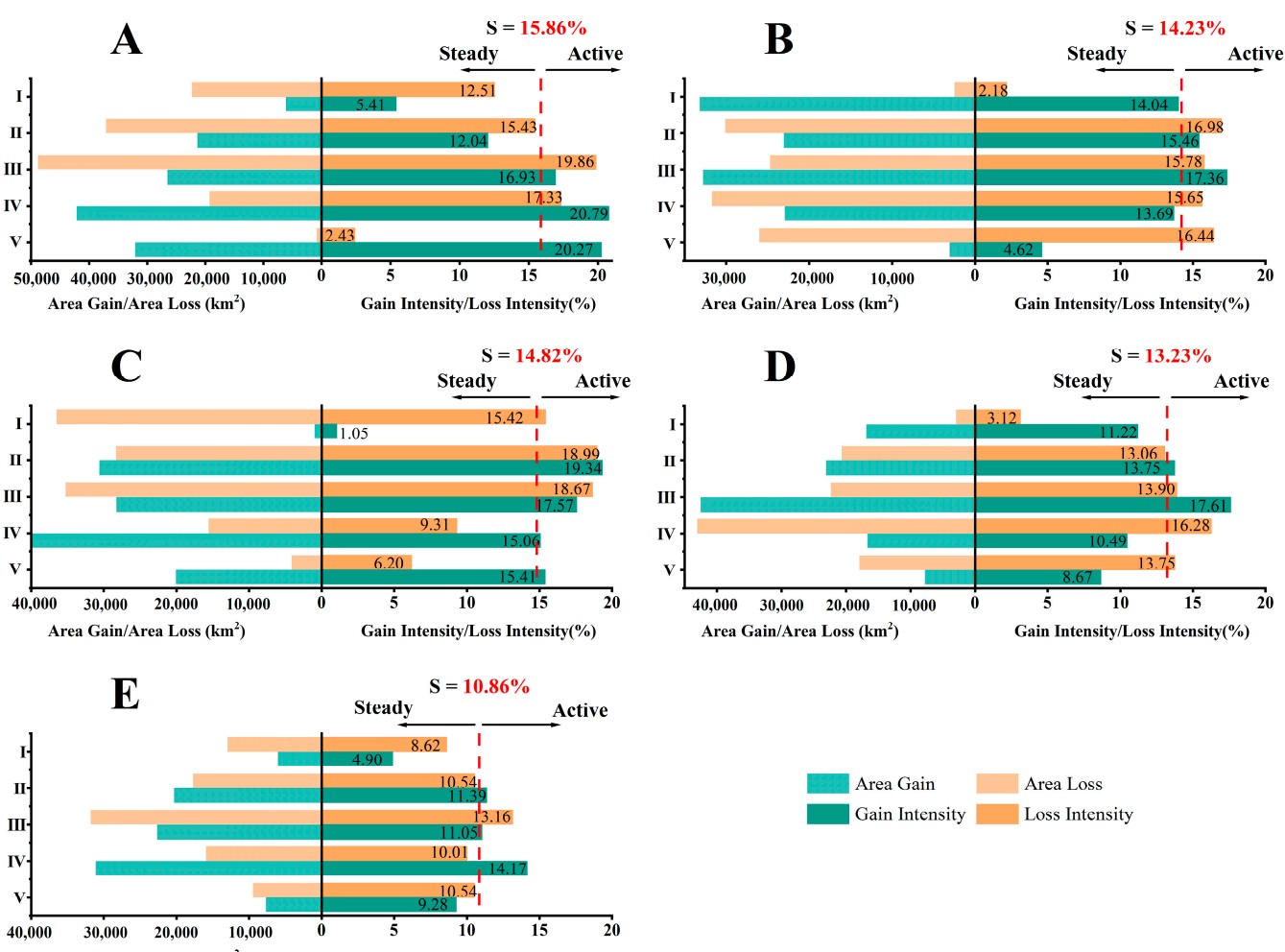

**Figure 8.** Category level of intensity analysis for (**A**) 2000–2004, (**B**) 2004–2008, (**C**) 2008–2012, (**D**) 2012–2016, and (**E**) 2016–2020.

In summary, the DGL categories exhibited acute changes in intensity during the 2000–2004, 2004–2008, and 2008–2012 periods. Desertification trends were positive during the 2000–2004 and 2008–2012 periods, but deteriorated during the 2004–2008 and 2012–2016 periods.

### 3.2.3. Transition Level

At the transition level, the study analyzed statistical data pertaining to the changes in grassland areas during a specific period (Figure 9). With Figure 9A taken as an example, values to the left of zero denote the proportion of the level I area that originated from levels II, III, IV, and V, while values to the right represent the portion of the level I area that transformed into levels II, III, IV, and V.

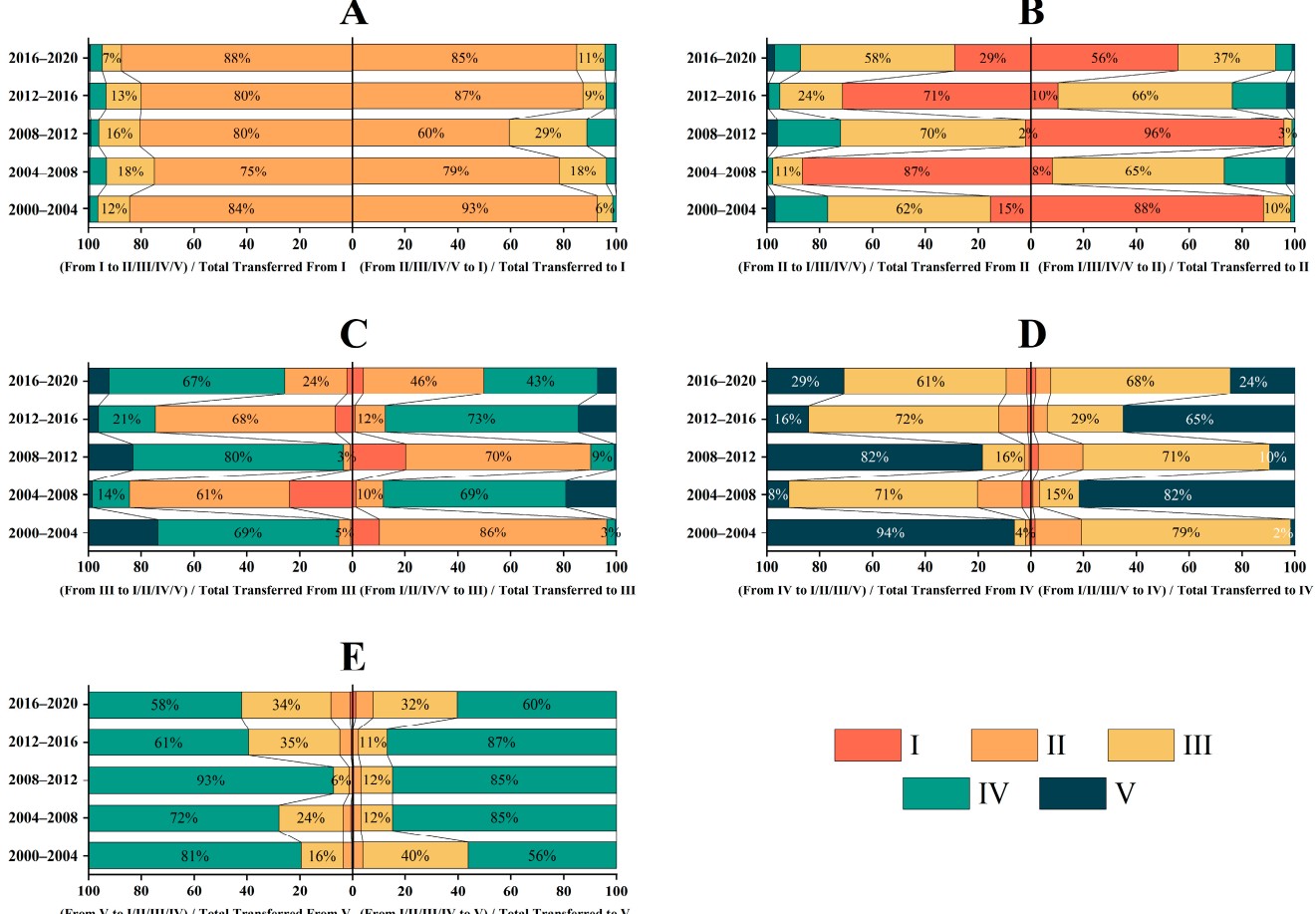

**Figure 9.** The sources of increased area and destinations of decreased area for different levels of DGL during all periods ((**A**) level I, (**B**) level II, (**C**) level III, (**D**) level IV (**E**), and level V).

Figure 9B indicates that during the 2004–2008 and 2012–2016 periods, among all areas transitioning out of level II, the majority shifted to level I. Simultaneously, in all areas transitioning into level II, the largest portion came from level III, which converted into level II. Similar trends can be observed in Figure 9C,D during the same time periods, signifying a consistent pattern of land transitioning from lower to higher desertification levels. This suggests an exacerbation of desertification trends during 2004–2008 and 2012–2016.

In terms of transition intensity (Figure 10), with Figure 10A taken as an example, values to the left of zero signify the intensity of transition from level I to other DGL levels, whereas values to the right indicate the intensity of transition from other levels to level I. The red dashed line to the left illustrates the average transition intensity from level I to other levels, and the corresponding line to the right represents the average transfer intensity from other levels to level I. Figure 10 presents a specific range of values, and values outside this range are annotated.

From Figure 10A, it can be observed that the intensity values of transitions from level I to level II and from level II to level I were the maximum values among all transition intensities involving level I. Specifically, during the 2000–2004, 2008–2012, and 2016–2020 periods, the intensity of transitions from level I to level II was greater than the intensity of transitions from level II to level I, indicating a decrease in the area of level I and an increase in the area of level II during these three periods. A similar situation is evident in Figure 10C, where the intensity of transitions from level III to level IV and level V is greater than the intensity of transitions from level IV and level V to level III, suggesting an increase in the areas of level IV and level V.

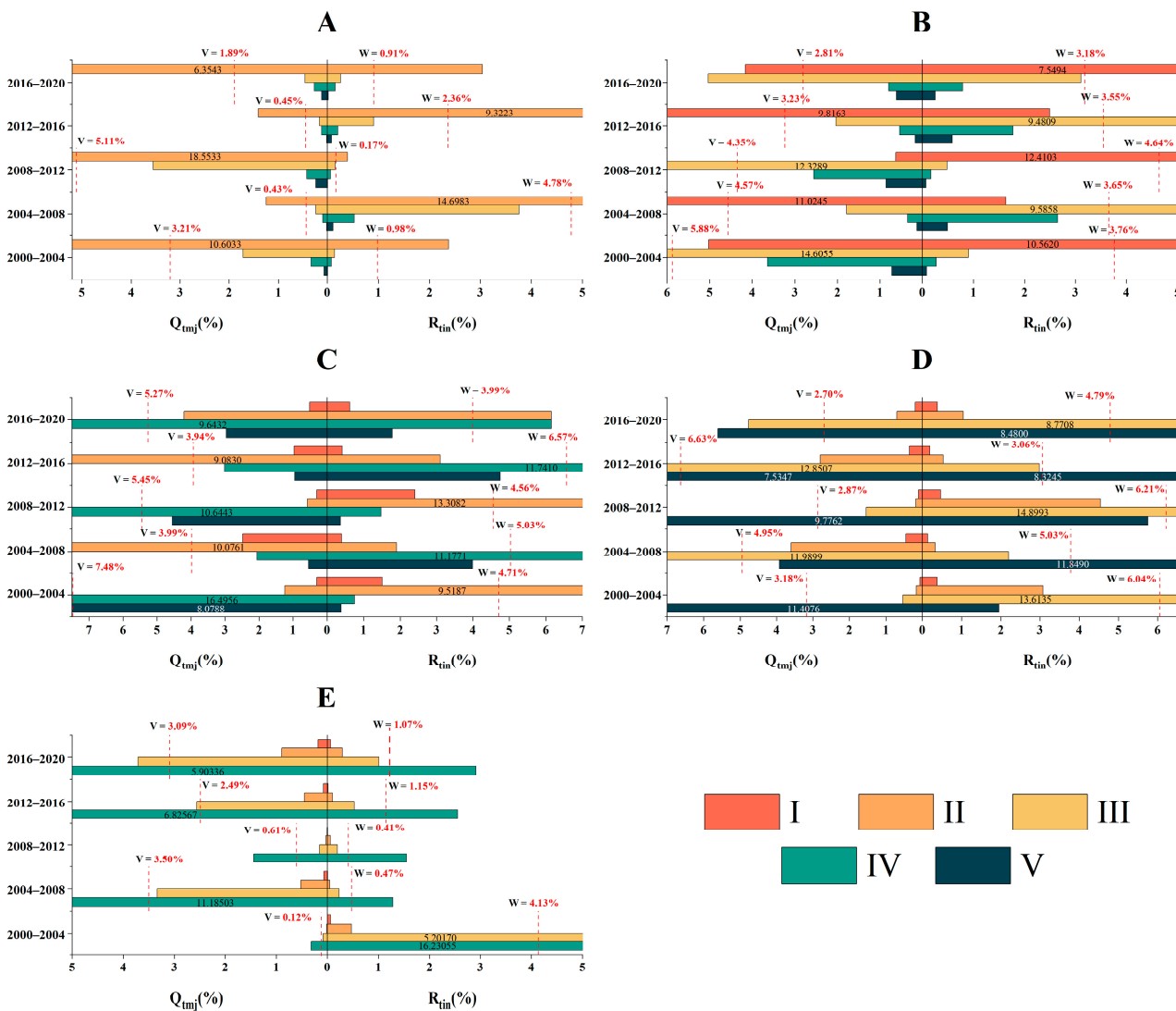

**Figure 10.** Transition level of intensity analysis ((**A**) level I, (**B**) level II, (**C**) level III, (**D**) level IV, and (**E**) level V).

In summary, there was an overall improvement in desertification trends during 2000–2004, 2008–2012, and 2016–2020. However, there was a certain degree of exacerbation of desertification during 2004–2008 and 2012–2016, with level III exhibiting the highest intensity of gain and loss. Notably, the transition intensities between adjacent levels consistently surpassed the average transfer intensity. This pattern underscores that grassland desertification is a gradual process, characterized by shifts between neighboring levels.

### 3.3. Driving Factors of Grassland Desertification

#### 3.3.1. Driving Factors of DDI Spatial Distribution

In the model inputs, the dependent variable (y) was represented by the calculated average value of the multiyear DDI. The independent variables (x) encompassed four categories of factors constituting a total of 14 factors: reanalysis meteorological data, soil data, topographic data, and human activity data (Figure 11). Notably, land surface temperature, precipitation, wind velocity, evapotranspiration, soil moisture, and population density were represented by average values spanning 2000 to 2020. Conversely, soil type, soil erosion type, soil erosion intensity, elevation, slope, and aspect were treated as single values, as these factors were not expected to undergo significant changes during the study

period. Geographic distribution data for livestock load employed the average values of currently available data, while land use and cover changes (LUCC) were based on the latest classification results from the IGBP (International Geosphere-Biosphere Programme) in the year 2020.

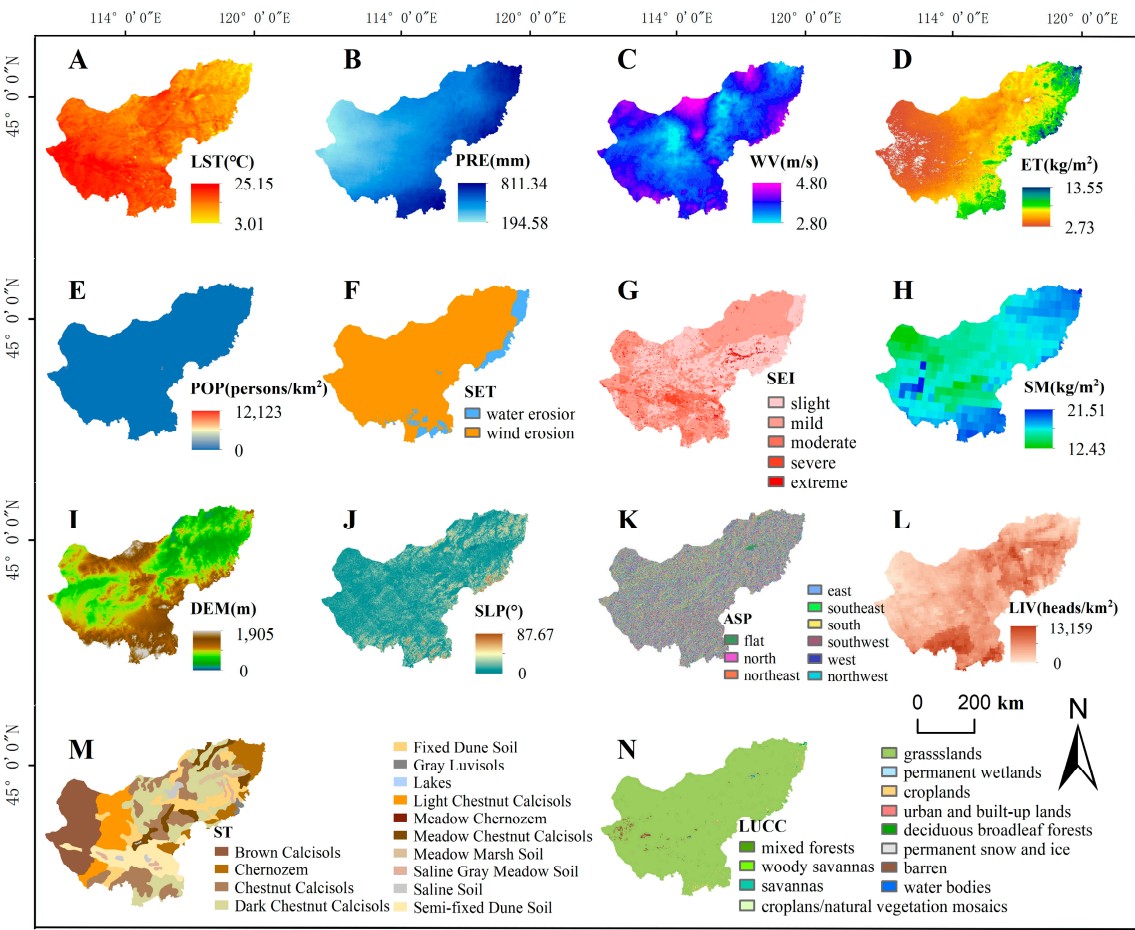

**Figure 11.** Input factors of the Geo-detector model. In the figure, (**A**): LST (Land Surface Temperature), (**B**): PRE (Precipitation), (**C**): WV (Wind Velocity), (**D**): ET (Evapotranspiration), (**E**): POP (Population), (**F**): SET (Soil Erosion Type), (**G**): SEI (Soil Erosion Intensity), (**H**): SM (Soil Moisture), (**I**): DEM (Elevation), (**J**): SLP (Slope), (**K**): ASP (Aspect), (**L**): LIV (Livestock), (**M**): ST (Soil Type), (**N**): LUCC (Land Use and Cover Change).

Within the study area, 3000 random points were generated, and numerical values for each factor and DDI were meticulously extracted at these points. Following thorough data validation, these values were input into the Geo-detector model for rigorous calculations.

In factor detector, it was observed that meteorological factors and soil factors, on the whole, exhibited significant explanatory power concerning the spatial distribution of DDI (Figure 12). Among these factors, evapotranspiration displayed the highest *q*-value, reaching 0.83, followed by precipitation, soil type, and land surface temperature, which also played crucial roles in driving DDI distribution. In contrast, aspect exhibited the weakest explanatory power.

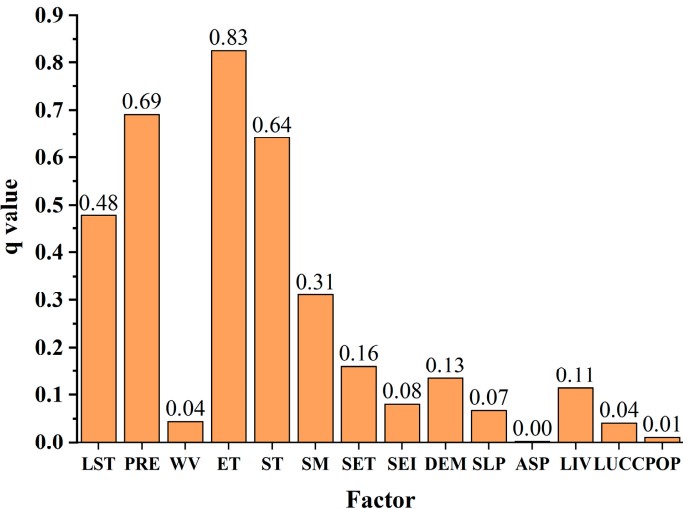

**Figure 12.** Results of factor detector.

Ecological detector is used to determine whether there are significant differences in the spatial distribution impacts of each pair of independent variables on the dependent variable. The results indicated that (Figure 13) concerning surface temperature, notable distinctions were present in its impact on the spatial distribution of the DDI when compared to precipitation, evapotranspiration, and soil type. Additionally, significant variations were observed in the influence of precipitation and evapotranspiration on the spatial distribution of DDI. In the context of wind velocity, there were significant differences in its impact on the spatial distribution of DDI in comparison to evapotranspiration, soil type, soil moisture, soil erosion type, elevation, and livestock. The effects on the spatial distribution of DDI also exhibited significant disparities for the soil erosion intensity–DEM, slope–livestock, and aspect–livestock factor pairs. For other factor pairs, there was no significant difference in their impact on the spatial distribution of DDI.

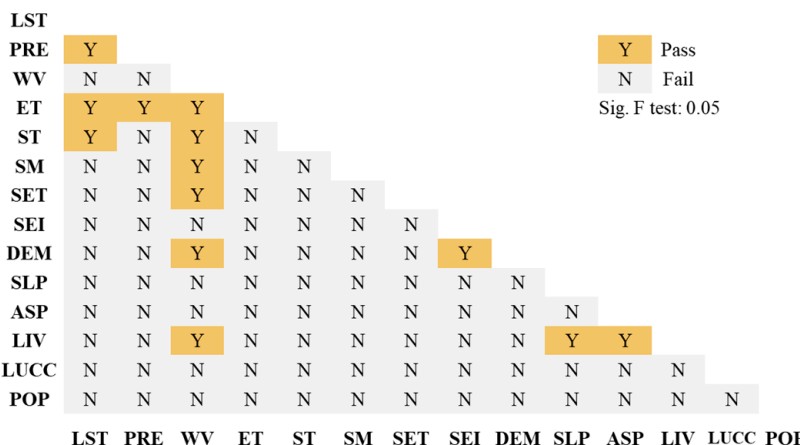

**Figure 13.** Results of ecological detector.

Interaction detector (Figure 14) indicated that the combined effect of any two factors in this study enhanced the explanatory power of the spatial distribution of DDI. Among them, the joint effect of evapotranspiration and soil type exhibited the strongest explanatory power on the spatial distribution of DDI, reaching 0.87, followed by the combined effect of evapotranspiration and land surface temperature, which reached 0.85. Overall, the combination of water, heat, and soil exerted a dominant driving role in the spatiotemporal distribution of desertification.

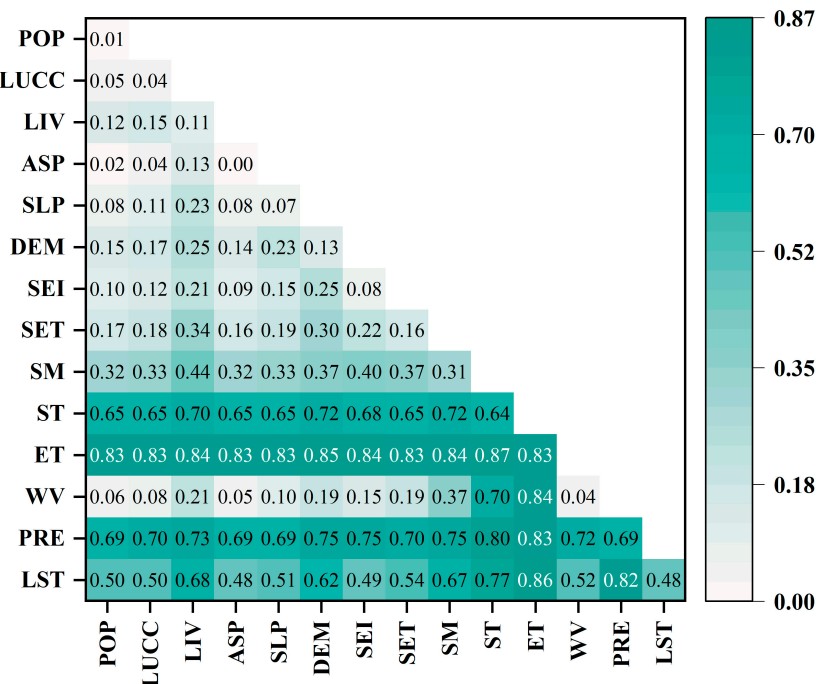

**Figure 14.** Results of interaction detector.

### 3.3.2. Driving Factors of DDI Temporal Variation

To explore the driving factors of DDI temporal variation, seven factors were selected and categorized into meteorological factors and human activities. Meteorological factors included annual average land surface temperature (LST), average annual precipitation, average wind velocity, and average evapotranspiration in Xilingol each year. Human activities included population, GDP, and livestock headcount (end of June). The study analyzed the numerical changes and trends of these factors from 2000 to 2020 (Figure 15). As depicted in Figure 15, the DDI mean value showed an increasing trend from 2000 to 2020, indicating a positive direction in desertification to some extent. Both population and GDP exhibited noticeable upward trends, while livestock headcount exhibited significant fluctuations with an overall decreasing trend, possibly due to local grazing control policies. The trends for LST and wind velocity changes were unclear, while precipitation and evapotranspiration showed significant increasing trends.

These two sets of data were then separately correlated with DDI to explore which set exerted a stronger driving force on the temporal variation in DDI. The results, as shown in Table 4, indicated that human activities exerted a stronger driving force on the temporal variation in DDI, with a significance level of 0.012, which is less than 0.05.

**Table 4.** Results of correlation analysis.

|  | Human Activities | Meteorological Factors |
|---|---|---|
| $\rho$ | 0.536 | 0.111 |
| Significance | 0.012 | 0.632 |

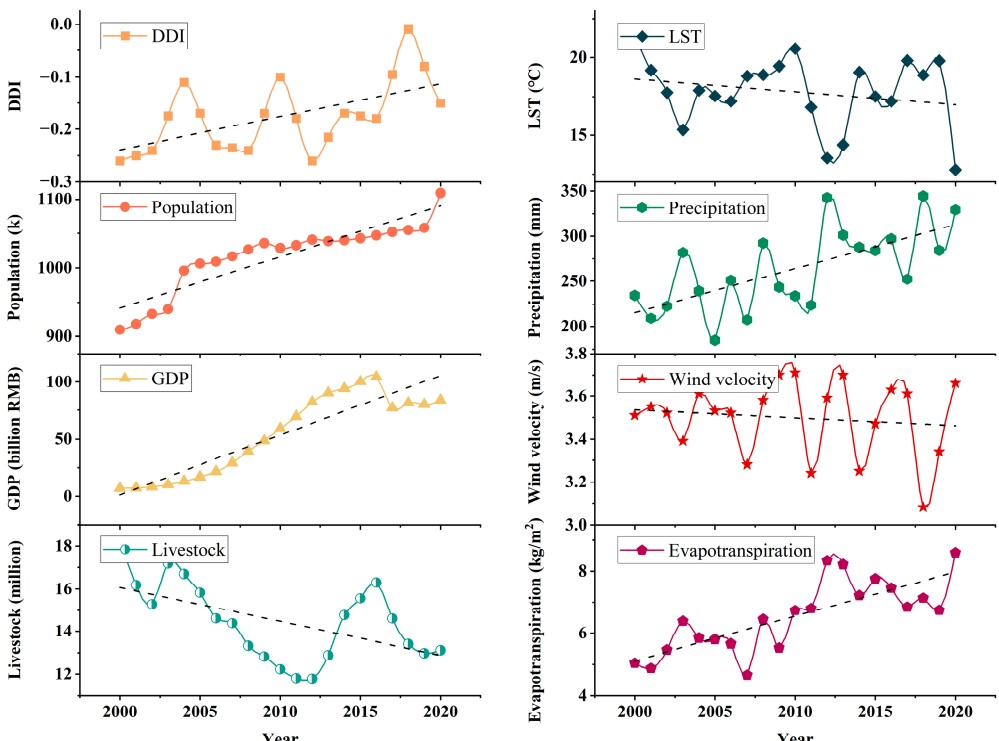

**Figure 15.** Temporal variation of factors from 2000 to 2020 (the black dashed line is the result of linear fitting).

## 4. Discussion

### 4.1. Dynamics of Grassland Desertification in Xilingol

This study utilized the albedo-EVI to construct a DDI model and conducted a spatiotemporal analysis of desertification dynamics on the Xilingol grassland. Furthermore, quantitative analysis of desertification intensity was performed using intensity analysis. These findings are of significant importance for understanding the trends in DGL variations in the Xilingol and for implementing effective desertification prevention measures.

Presently, research employing the albedo-EVI model for DDI construction is limited. Existing findings indicate that NDVI exhibits saturation in regions with high vegetation [52], impeding the model's ability to accurately discern actual vegetation information in these areas and consequently diminishing its capacity to precisely evaluate desertification degrees. With an increase in vegetation cover, particularly in moderately desertified areas, NDVI is subject to soil factors, resulting in the oversight of some newly added vegetation information [53]. This oversight leads to misclassification, wherein the model erroneously identifies moderately desertified areas as severely desertified. Consequently, in comparison to the albedo-NDVI model, the albedo-EVI model demonstrates enhanced accuracy in extracting desertification information.

From Figures 6 and S1 and Table S1, it can be seen that desertification degree on the Xilingol grassland improved between 2000 and 2020. This finding aligns with the results reported by Batunacun et al. and Li et al. [54,55]. However, due to differences in the selected indices and threshold delineation used to characterize the degree of desertification, the classification results exhibit variations. For instance, some studies utilize NDVI values or vegetation coverage to represent land degradation or the degree of desertification [26,40,43,44]. It is essential to clarify that the primary focus of these studies is not on achieving precise classification of desertified land but rather on reflecting the spatiotemporal trends of desertification, and this study shares a similar focus. Furthermore, there are currently no established national or industry standards for the classification of lands with different levels of desertification. As a result, this study did not delve into an in-depth investigation of desertification level classification.

The intensity analysis results show that at the interval level, dynamic changes in different levels of DGL have gradually stabilized. At the category level, conversions among different levels of DGL were particularly intense during 2004–2008, indicating an increasing trend in grassland desertification. Transitions at the desertification level mainly occurred between adjacent levels, with substantial year-to-year variations, possibly attributed to meteorological conditions. In the context of global climate change and frequent extreme weather events, local microclimates in Xilingol are significantly influenced, resulting in unstable annual precipitation [56]. The grassland's responsiveness to water availability induces rapid improvements in grassland quality following rainfall, consequently reducing the degree of desertification.

*4.2. Driving Factors of Grassland Desertification in Xilingol*

Geodetector and correlation analyses were used to comprehensively explore the driving factors influencing spatial and temporal variations in desertification on the Xilingol grassland. Unlike some prior studies that focused on specific factors, this research systematically classified driving factors into meteorological, soil, topographical, and human activity categories. Surface soil moisture and soil erosion factors, often overlooked in previous studies, were found to impact the spatial distribution of desertification. Additionally, livestock data included goats, sheep, cattle, and horses, recognizing their significant influence on grassland quality, thereby enhancing the scientific rigor of the analysis.

Spatially, the factor detector results showed that evapotranspiration was the most significant driving factor for the spatial distribution in desertification on the Xilingol grassland, which aligns with the results of Han et al. [28]. Since most studies did not consider ET as one of the influencing factors in the Geo-detector model, there are limited experimental results available for comparison and validation. However, evapotranspiration is strongly correlated with temperature and precipitation. In this study, precipitation, soil type, and land surface temperature were also major driving factors for DDI spatial distribution, which is consistent with the results of Wang et al., Xu et al., and Zhu et al. [25,40,44].

Precipitation, soil type, and land surface temperature emerged as pivotal factors influencing desertification. Insufficient precipitation induces soil aridity, impeding vegetation growth, escalating soil wind erosion and erosion, and intensifying desertification [57,58]. Different soil types exhibit varying water retention and wind erosion resistance capacities, influencing the rate of desertification [58,59]. Elevated land surface temperatures accelerate soil moisture evaporation, adversely affecting vegetation growth, and potentially triggering meteorological and climatic changes that exacerbate desertification trends [59,60]. The interplay of these three factors collectively contributes to vegetation degradation and soil impoverishment, accelerating the progression of desertification. A comprehensive consideration of these factors is crucial for scientifically preventing and mitigating desertification.

Meteorological factors, particularly evapotranspiration and precipitation, were identified as the most significant driving factors for spatial distribution differences in grassland desertification. Figure 11B illustrates minimal precipitation in the western part of Xilingol, indicating more severe drought conditions compared to the eastern areas. Due to water scarcity, the western region experiences poorer vegetation growth, resulting in lower vegetation cover and evapotranspiration, thereby leading to more severe desertification.

In terms of temporal variation, Spearman correlation analysis results indicated that human activities were the main driving factors for temporal changes in desertification on the Xilingol grassland. Similar conclusions were drawn by Li and Xie [36], emphasizing the crucial role of human activities in temporal desertification variation. The impact of human activities could be positive or negative. Figure 15 shows that from 2000 to 2020, both the population and GDP in the Xilingol region exhibited an overall upward trend, suggesting intensified land use and potential land desertification. However, the urbanization rate in Xilingol increased from 35.60% in 2000 to 73.88% in 2020. This indicates that during this period, a significant number of herders moved to cities, potentially reducing

human interference with the grassland and contributing to the alleviation of grassland desertification. This aspect requires further investigation.

In recent years, China has actively promoted various ecological projects, including the project of "Integrated management of mountains, rivers, forests, farmlands, lakes, grasslands, and deserts". The importance of grassland management has been emphasized in governmental documents, in which grasslands are described as the "skin of the earth" [61]. Xilingol is a vital ecological conservation area in northern China. Ecological condition of the grassland has gradually improved with the efforts of the official and public. This improvement in grassland quality aligns with the results obtained from intensity analysis.

### 4.3. Limitations and Future Work

This study has certain limitations that should be acknowledged. In terms of data selection, despite various driving factors being considered, comprehensiveness remains an issue. Notably, policy factors, such as rotational grazing, forbidden grazing, or grazing suspension, which can significantly impact grassland desertification in Xilingol, were not included due to challenges in quantification and incorporation into spatial distribution data suitable for the Geo-detector analysis.

Data utilization, particularly regarding livestock data, poses limitations, as the majority of such data were sourced from statistical yearbooks lacking rasterized geographic data products. Given the spatially varying distributions required for geographic detectors, this study relied on mean values from only two obtainable years, potentially influencing the results.

In the analysis of spatial changes and driving forces, the human activity factors derived from statistical yearbook data lacked uniformity. Although justified for certain reasons, such as limitations in data format and accuracy concerns, these uncertainties may impact the results and represent an area for improvement in future research.

Additionally, uncertainty exists in the method of threshold division. The absence of industry standards for defining desertification levels leads to variations in results as scholars define thresholds based on their own research. Addressing this issue is essential in future work.

The validation of desertification classification in this study relied on visual comparison using Google historical imagery, a method subject to subjectivity. Future research should consider validation through more objective approaches such as field investigations.

In future work, addressing the limitations of the current study requires comprehensive efforts. First, there is a need to incorporate policy factors, such as rotational grazing and grazing suspension, into spatial distribution data to better understand their impact on grassland desertification. Second, efforts should be directed toward obtaining or developing rasterized geographic data products to enhance the accuracy and reliability of livestock data. Third, exploring alternative sources or methodologies for deriving human activity factors can address the lack of uniformity in data. Establishing industry standards for desertification level definition is crucial for ensuring consistency across studies. Lastly, improving the validation process of desertification classification through more objective methods, such as field investigations or remote sensing techniques, will contribute to the overall robustness of future research in this domain.

### 4.4. Contributions and Significance

Through quantifying variations in desertified grassland (DGL) area, analyzing spatial patterns, and elucidating conversion processes among different desertification levels, this study provides crucial insights into the dynamics of desertification in both temporal and spatial dimensions. These findings offer valuable information for targeted interventions in environmental protection, aiding in the prevention and control of desertification in Xilingol by identifying areas with varying levels of desertification and understanding the intensity and patterns of conversion. The study's emphasis on meteorological factors, soil conditions, and human activities as driving forces underscores the importance of adopting

sustainable land management practices. Regarding climate change, the identification of evapotranspiration as a significant driver highlights the interconnectedness of land-use changes and climate dynamics [62]. This research contributes to the broader discourse on environmental sustainability and climate resilience, providing a foundation for evidence-based policies and practices to address grassland desertification amid global environmental challenges.

## 5. Conclusions

This study systematically examined the spatiotemporal dynamics of the Desertification Difference Index (DDI) in Xilingol over the 2000–2020 period. The investigation employed intensity assessments on grassland exhibiting diverse levels of desertification, considering intervals, categories, and transition levels. Geodetector and correlation analyses were applied to elucidate the driving forces behind the spatial distribution and temporal variation of DDI. The results indicated that the overall degree of desertification in Xilingol showed an improvement from 2000 to 2020.

Analysis revealed a discernible zonal pattern in the spatial distribution of DDI, exhibiting a gradual decrease from east to west, aligning with the distinct distribution of grassland types in Xilingol—specifically, meadow steppe, typical steppe, and desert steppe. Over the study duration, DDI demonstrated a consistent increasing trend, corresponding to a reduction in the extent of level I and level II areas, juxtaposed with an expansion in areas characterized as level IV and level V. Notably, the proportion of level I and level II decreases from 51.77% in 2000 to 37.23% in 2020, while the proportion of level IV and level V increased from 17.85% in 2000 to 37.40% in 2020. However, it is important to mention that the area corresponding to level V underwent a reduction during the 2000–2020 period.

The intensity analysis further discerned noteworthy trends, indicating rapid changes in DDI from 2000 to 2012, followed by a deceleration from 2012 to 2020. At the category level, the conversion intensity for level III, level IV, and level V reached its pinnacle from 2008 to 2012. In the transition level analysis, level III exhibited the highest intensity of both gain and loss.

The study emphasizes the influential role of meteorological factors and soil conditions in shaping the spatial distribution of DDI. Notably, evapotranspiration emerged as the dominant factor, with the highest $q$-value at 0.83, followed by precipitation, soil type, and land surface temperature. In contrast, temporal variations were predominantly attributable to human activities.

In conclusion, this research significantly contributes to advancing our understanding of the spatiotemporal dynamics of desertification in Xilingol. It offers valuable insights into the conversion patterns among different levels of desertification within Degraded Grassland (DGL), elucidating the distinct driving factors influencing the spatial distribution and temporal variation of desertification. The implications of these findings extend significantly to ecological research and conservation efforts in the region.

**Supplementary Materials:** The following supporting information can be downloaded at https://www.mdpi.com/article/10.3390/rs15245716/s1, Figure S1: Proportions of DGL at different desertification levels; Table S1: Statistics of areas with different desertification levels (unit: $km^2$).

**Author Contributions:** Conceptualization, J.L. and M.X.; data curation, X.Y.; formal analysis, J.L.; funding acquisition, M.X.; investigation, M.X., K.W. and Y.Y.; methodology, J.L. and X.Y.; project administration, C.C.; resources, X.G.; software, H.G.; supervision, C.C.; validation, X.G. and K.W.; visualization, H.G. and Y.Y.; writing—original draft, J.L.; writing—review and editing, C.C. All authors have read and agreed to the published version of the manuscript.

**Funding:** This work was supported by the project of the National Key R&D Program of China (grant number 2021YFB3901104) and the National Natural Science Foundation of China (grant number 41971394).

**Data Availability Statement:** Data are contained within the article and supplementary materials.

**Acknowledgments:** We would like to thank the Resource and Environmental Science and Data Center, Chinese Academy of Sciences, for data support. We also extend our sincere gratitude to Cao and Xu for their guidance on the structure and language of the paper.

**Conflicts of Interest:** The authors declare no conflict of interest.

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
