# Peer review of "A 20-Year Analysis of the Dynamics and Driving Factors of Grassland Desertification in Xilingol, China"

_remotesensing, doi:10.3390/rs15245716_

Round 1
Reviewer 1 Report
Comments and Suggestions for Authors
1. Introduction: Please clarify the novelty of this research compared with previous studies and explain the research significance from the perspective of global interests instead of focusing on local interests of the case study area Xilingol.
2. Table 4 and Figure 5 are duplicated. I would suggest deleting Table 4 or moving it to the supplementary materials.
3. Lines 321 to 322: Please explain using a period of four years to divide the study period into five sub-periods.
4. Lines 432 to 435: Please explain the assumptions behind the average weighting calculation.
5. The present research lacks contribution to remote sensing technology.
Reviewer 2 Report
Comments and Suggestions for Authors
Even though this is an interesting study, it could be useful to understand desertification for any particular area. However, the article needs some revisions. Please address the comments to improve the quality of your article.
1. Abstract must be concise. Probably remove points from Abstract.
2. The introduction also needs improvement. For example, in the introduction line 67-68, authors stated that "...NDVI has become one of the most...". I suggest that the authors provide the following reference to support the argument: Sarker, Shiblu, "Investigating Topologic and Geometric Properties of Synthetic and Natural River Networks under Changing Climate" (2021). Electronic Theses and Dissertations, 2020-. 965. https://stars.library.ucf.edu/etd2020/965
3. Figures 1 require revision. On the map of their study area, authors may include DEM, river networks and systems. Please review the manual for ArcGIS or another professional software in order to generate publication-quality figures. Additionally, figures 3, and 10 need substantial revision. Again, review the ArcGIS manual in order to produce figures suitable for publication.
4. I am not convinced by the writing in Methods! Currently, it is poorly written! Please review additional articles to improve your methodology. Also can you add a nice and clean flow chart?
5. What is the R^2 value of your figure 2? Is this good enough? Probably you can do a t-test to justify this. See the aforementioned reference and cite that reference the t-test analysis.
6. Replace figure 4 with a nice heatmap. You may use Python or Matlab to generate publishable figures. Please review this python toolbox. https://timcera.bitbucket.io/plottoolbox/docs/index.html. Figures 5, 6, 7, 8, 9, 11, 12, and 13 should also be revised. If feasible, combine all factors into a single figure for figure 14, applying two y-axes or reduce the space between subplots.
7. What is the significance of this research? Please explain this study's implications in terms of environmental protection and climate change. Please describe the potential implications of this study in a separate section (prior to the conclusion). It does not currently convince me that it provides a foundation for environmental protection and climate change. Please review the following literature to address this issue. Sarker, Shiblu. "Fundamentals of Climatology for Engineers: Lecture Note." Eng 3.4 (2022): 573-595.
Comments on the Quality of English LanguagePerhaps it would be beneficial for the authors to revise their compositions, particularly the sentence structure.
Reviewer 3 Report
Comments and Suggestions for Authors
The current advancements in desertification monitoring methods are exceptionally rare. The author's enhancement of NDVI using EVI represents progress. Additionally, the combination of intensity analysis and geographic detectors for extensive quantitative analysis is also a step forward. Overall, the research design is reasonably sound, with a certain level of methodological innovation, but there is still a need for improvement and further reflection. I hope the following suggestions can help the author enhance the paper's quality.
1.Introduction
The introduction fails to highlight the innovation of this study.
(1) The author only elaborates on the advantages of using vegetation indices for desertification monitoring, failing to summarize the mainstream methods in the current remote sensing domain of desertification and the strengths of the methods employed in this study. It is essential to provide a focused introduction to the methods used in the paper and deduce the innovations of this research from there.
(2) Similarly, the author does not effectively justify the application of intensity analysis and geographic detectors in the field of desertification. The narrative lacks logical coherence.
(3) The aforementioned reasons result in a lack of some relevant literature in this study.
2. Data
The section on data sources and processing lacks essential information.
(1) The entire paragraph emphasizes the source and type of data, which is repetitive with Table 1.
(2) Line 151: The author utilizes reanalysis data rather than meteorological data. Therefore, precision is needed in the expression.
(3) For desertification monitoring, the selection of months for Albedo and EVI is crucial. Using images from pre-greening and senescent periods for monitoring can result in significant errors.
(4) Each dataset lacks attributes such as spatial and temporal resolution. How is the resolution standardized between different spatial resolutions?
(5) The study primarily relies on reanalysis data. What is the applicability of these data in the study area, and what criteria were used for their selection?
3. Results
The results section lacks logical coherence, depth, and critical discussion of the significance of the presented data.
(1) The description of desertification monitoring results in section 3.1 is confusing. Table 4, Figure 4, and Figure 5 need to be discussed together. What are the main findings derived from the monitoring data? Has desertification progressed or reversed? What is the predominant level of desertification in the study area over different time periods? How do lands of different desertification levels change? Table 4 and Figure 4 might be better suited as supporting documents, using change values instead.
(2) The author's classification method is problematic since the non-desertified land area across the entire study area is very small, which is evidently unrealistic. This issue needs to be addressed through on-site investigations or high-resolution satellite imagery from sources like Google Earth. If the author does not accept this suggestion, there should be a thorough comparative analysis of the desertification monitoring results in the discussion section, providing a reasoned discussion on the classification method.
(3) The use of abbreviations in the text makes it challenging to read. In the intensity analysis, it needs clarification as to which period the changes in desertified land are most active, which level of land change is most active, and which changes between levels are most active. The overall analysis lacks depth, focusing more on explaining the methods, while the description and critical interpretation of the research results are insufficient. Additionally, there is a lack of effective integration of the data presented in the figures.
(4) The selected driving force data includes some annual values, some multi-year values, and some non-quantitative values. What are the dependent and independent variable data input into the model? This information is missing.
(5) The author chooses land surface temperature as a driving factor and ignores air temperature. What is the basis for this choice? Many studies have highlighted the significant impact of air temperature on desertification dynamics.
(6) The overall analysis lacks depth. For instance, in factor detection, which factor dominates, natural or anthropogenic? Static or dynamic? In interactive factors, is there evidence that the combination of natural and anthropogenic factors accelerates the desertification process? Or the combination of water and heat factors? What is the type resulting from the combination shown in Figure 3? What does the ecological detection's results indicate?
(7) Regarding temporal changes, the increase in DGP due to driving factors is perplexing. This should align with the factors in geographic detectors (excluding factors with only one-year values) rather than selecting new indicators. Coupled analysis should be conducted considering both desertification monitoring results and geographic detector results. For example, the author detects a high Q value for precipitation, and in Figure 14, precipitation is increasing, and non-desertified land is also increasing. Is there a connection?
(8) It is perplexing that Table 5 indicates a greater impact of anthropogenic factors than climatic factors, which is apparently contrary to the results of geographic detectors. The study needs to clearly identify the dominant factors driving desertification changes in the study area. Has the author considered these issues? If the author does not accept this suggestion, please provide a clear explanation of why the factor significantly affecting spatially becomes weaker over time. It is not evident from Figure 10 that meteorological factors determine the east-west spatial distribution pattern of desertification. Also, the author uses different data sources for livestock numbers and population in space and time. Can the conclusion be directly drawn that the spatial pattern is determined by meteorological factors? Furthermore, meteorological factors are dynamic and change rapidly. Additionally, in Figure 14, DDI is shown to be increasing, contrary to the results in Figure 5. The author needs to review and clarify the entire manuscript.
4. Discussion
(1) In section 4.1, there is a need to incorporate comparative analyses of desertification monitoring results. Are the results of this study consistent with previous research? If not, what are the reasons? What are the advantages of the monitoring methods employed in this paper? At least a comparison between Albedo-EVI and Albedo-NDVI methods should be made to demonstrate the applicability of the Albedo-EVI method in grassland desertification monitoring.
(2) The content from Line 459-465 should be included in section 4.2. First, discuss the driving factors considered in this study and then introduce the factors not considered. Further explanations of the results are needed, such as the high impact of precipitation, soil type, and LST on desertification. Why do they influence desertification, and how? The author claims that others have not considered evaporation, which is evidently incorrect. Has evaporation's relationship with drought not been considered in other studies? The author should consult relevant literature to understand the connection between evaporation and desertification.
(3) I believe the limitations of this study include not only the quantification of policy factors but also the absence of data, classification methods, and fieldwork.
5. Conclusion
The conclusion section needs to be reorganized, and it is recommended not to use abbreviations in this part.
(1) From the research results, it is evident that the area of V-level is decreasing, and high-level desertified land is also decreasing. This indicates a reversal of desertification in the study area, with a decrease in levels. A comprehensive summary is needed, considering key data.
(2) The author mentions that the spatial distribution of desertified land corresponds well with the distribution of vegetation types, but this finding is not mentioned in the results section.
(3) The conclusion regarding changes between adjacent levels should be reevaluated for its value. The author needs to extract more useful information from the results section.
(4) It is crucial to identify the dominant factors driving desertification in the study area and integrate key data such as Q values.
Round 2
Reviewer 2 Report
Comments and Suggestions for Authors
Thanks for the revision.
Comments on the Quality of English LanguagePlease check again.
Reviewer 3 Report
Comments and Suggestions for Authors
The revised manuscript shows significant improvement in the results analysis and discussion section, but there are still some issues with the overall revised manuscript. Please provide a clear version after the author's modifications, with the changes displayed in track changes mode.
1.Introduction
The author can restructure the introduction in the following order to achieve clear logical flow. First, please merge the first and second paragraphs. The fourth paragraph can be deleted. Next, follow the third paragraph (Lines 72-82), followed by the sixth paragraph (Lines 110-124) and the seventh paragraph (Lines 125-136). Subsequently, include the fourth paragraph (Lines 99-109), followed by the eighth paragraph (Lines 137-149). Finally, state the research objectives of this paper.
2.Data
I understand the author's explanation regarding the reanalyzed data. However, the author needs to add a statement at the end, informing readers that these data have been well applied in the study area or the Chinese region, along with relevant references.
3.Methods
Why did the author delete the original Figure 2? Was there an operational error? Please retain the figure and add information on the model's significance or R-squared value. This is crucial to demonstrate the validity of the methods used.
4.Results
(1) Please delete Lines 401-406.
(2) The abbreviation for 'land use and land cover' is LULC; please verify.
(3) What is 'IGBP'?
(4) I can understand the trend of DDI in Figure 14. Please explain in Section 2.3.1 what DDI represents. For example, a high DDI value indicates a lower degree of desertification; otherwise, the wording in the manuscript is not coherent.
(5) Please move Lines 698-705 to the methods section.
5.Conclusion
I believe the author's writing in this section is still poor. The first paragraph does not inform readers whether desertification in the study area has progressed or reversed.
